# Female sexual agency and frequent extra-pair copulations, but no extra-pair paternity, in Nazca boobies (*Sula granti*)

**David J. Anderson** [1]*, **Kathryn P. Huyvaert**[2], **Paola M. Espinosa**[1,3], **Mark A. Westbrock**[1], **Jill A. Awkerman**[1], **Eric G. Schneider**[1]

**1** Department of Biology, Wake Forest University, Winston-Salem North Carolina, United States of America, **2** Department of Veterinary Microbiology and Pathology, Washington State University, Pullman Washington United States of America, **3** Departamento de Ciencias Biológicas, Pontificia Universidad Católica, Quito Ecuador

* djanders@wfu.edu

## Abstract

Extra-pair copulations (EPCs) are the poorly known antecedents of extra-pair fertilizations (EPFs) in birds. EPFs occur in most bird species that have been examined, but sexual conflict will generally reward females hiding their EPCs from males attempting to protect their paternity. EPCs will be difficult for researchers to document, and necessarily underestimated, in that case. We measured EPC behaviors and EPF frequency in a colonial seabird, the Nazca booby *Sula granti*, in which all copulations occur in a visually open setting with numerous possible copulatory partners readily available. Females are larger and more physically powerful than males, and are the numerically limiting sex, perhaps limiting options for males to control females. We found that all copulations were voluntary, and females' sexual activities were wholly unconstrained by male coercion. Most females had multiple copulatory partners in the weeks preceding egg-laying. Despite the commonness of EPC, EPFs did not occur. The different schedules of EPC and within-pair copulation (WPC) provided a sufficient explanation for this outcome: during the ovulation window days before laying, WPC rate increased and EPC rate approached zero. To our knowledge, this is the first robust evidence of complete sexual agency in a female bird aside from lek-mating species, contributing a valuable exemplar to the literature on sexual conflict over reproduction.

## Introduction

Extra-pair paternity (EPP) in broods of offspring is widespread among bird species [1]. Our understanding of EPP has grown dramatically with the application of molecular techniques to species across the Aves (e.g., [2]), while the extra-pair copulations (EPCs) that must precede them are less well-studied. Females in breeding pair

**Data availability statement:** All four files are available from the WakeSpace database (accession number https://wakespace.lib.wfu.edu/handle/10339/110830).

**Funding:** This material is based upon work supported primarily by the National Science Foundation (www.nsf.gov) under Grant No. DEB986–06606 to DJA. The funders did not play any role in the study design, data collection and analysis, decision to publish, or preparation of the manuscript.

**Competing interests:** The authors have declared that no competing interests exist.

bonds often have incentives to conceal their EPCs from their within-pair mate, to escape mate-guarding and/or to avoid assessment by him that EPC has occurred and the resulting consequences associated with his protecting his paternity of off-spring, avoiding communicable diseases, and other costs (e.g., [3]). Covert copulatory behavior to thwart male mates (e.g., [4]) in this sexual conflict may obscure EPC events from investigators, despite the commonness that EPP indicates, hindering the documentation of behaviors, costs, and benefits of EPCs. An exception to this statement is the lek mating system, in which females of at least some species appear to have complete sexual agency, and copulations are not covert [5]. For most non-lek species, covert EPCs prevent answering even fundamental questions: how much female promiscuity occurs in a given species, and which part of that is voluntary, and perhaps adaptive, on the female's part?

Some colonial birds provide opportunities to observe females in high-density assemblages and visually open settings. In the best cases, complete histories of EPCs and within-pair copulations (WPCs) can be collected from focal females if all copulations are on-site. Considering colonial seabirds, more than half of the species studied have EPP [6,7], and some behavioral work has illuminated how the EPCs occur, or are prevented, and tested hypotheses for their causation (e.g., [8–12]). Nonetheless, the potential to document female copulatory breadth — that is, their behavior as distinct from genetic outcomes — in colonial seabirds is largely untapped, and both EPCs (social behavior) and EPP (genetic mating system) have been documented in only a few species (waved albatross *Phoebastria irrorata* [10], little auk *Alle alle* [11], whiskered tern *Chlidonias hybrida* [13], blue-footed booby *Sula nebouxii* [14,15]).

The mating system of Nazca boobies (*Sula granti*), a tropical colonial seabird, implies female control over sexual interactions. Our long-term study population of Nazca boobies in the Galápagos Islands has seasonal, relatively synchronous annual breeding, obligate biparental care, and frequent divorce between breeding seasons (i.e., serial monogamy [16]). Females are substantially larger (13% by mass, on average) than males are [17], have stronger bite force [18], and show the social dominance over males [16,19] that often accompanies larger body size [20]. Females have lower post-fledging survival than males, due to parental provisioning in poor food years not fully meeting the nutritional requirements of daughters, the larger sex [21,22]. This excess female mortality produces a male-biased adult sex ratio among recruits to the breeding population [23]. As the limiting sex, females drive serial monogamy by often rejecting former mates [19] after successful breeding in favor of males that have not incurred recent costs of reproduction [16]. Because the excess ~1/3 of males cannot obtain a mate in a given breeding season, males may be motivated to accept any female that might pair with him, and tolerate whatever behavior she engages in. The median distance between nest sites in the flat, rocky, unvegetated colony is only 2 m [24], giving females abundant opportunities for visual contact with neighboring males and walking visits to them. All of these circumstances should facilitate females' sexual agency.

Casual observations of pre-breeding females indicated that some had multiple copulatory partners (DA, pers. obs.), motivating a molecular analysis of paternity of hatched young that estimated the probability of EPP as 0 (95% CI 0–0.10; [25]). Here, we characterize the sexual behavior of female Nazca boobies in a quantitative observational study. First, we demonstrated that EPCs (see Methods for operational definition) are common, an outcome perhaps inconsistent with the paternity result [25]. To enhance confidence in that estimate of EPP frequency, we conducted a second molecular parentage analysis on different samples collected during the present observational study, 12 years after the earlier [25] study. Returning to copulation behavior, we tested the hypothesis that females engage in within-pair copulations (WPCs) and EPCs voluntarily; that is, we evaluated the sexual agency of females and the frequency of apparently forced copulations. To reconcile the failure of frequent voluntary EPCs to produce extra-pair offspring, we compared the schedules of WPCs and EPCs in relation to egg laying, predicting that WPCs predominated in the 4 days before egg laying during which the fertilization window in this species must occur [26]. We also evaluated the possibility that EPFs do occur but are filtered from our dataset by selective male abandonment before blood sampling of the putative extra-pair offspring at hatching.

## Materials and methods

### Study system

This observational study was conducted within the breeding colony of approximately 10,000 Nazca boobies at Punta Cevallos, Isla Española, Galápagos Islands, Ecuador (1°23'S, 89°37'W), in conjunction with other long-term work on this species. All methods were approved in advance by the Wake Forest University Animal Care and Use Committee. These birds nest on open, unvegetated ground among rocks along the coast, within 35 horizontal m of the high tide line [24]. Our intensive daily monitoring for the previous nine years, and the species' natural tolerance of human presence, allowed data collection and capture for leg-banding or blood sampling with little disturbance. At Punta Cevallos most eggs are laid from October to January. Adults associate for weeks or months before egg laying, mostly at identifiable nest sites, attending the colony on roughly 50% of nights [27] and many of the intervening days. Only one offspring is raised per breeding attempt, but two-egg clutches are common, providing an insurance egg to counter low hatching success [28,29]. Some clutches produce two hatchlings, although obligate siblicide reduces the brood to one within days of the second egg's hatching [29,30]. Thus, each clutch involves one or two fertilization opportunities. The incubation period is 43–50 d, depending on clutch size and hatching success, and post-hatching parental care requires 5–6 months [31].

This study took place on a 266 m$^2$ (14 m x 19 m) focal plot (hereafter, "plot") at the northern end of Subcolony 1 [32], bounded on three sides by non-colony habitat and on the fourth by a gap separating it from the next part of the colony. We made observations from the top of an adjacent promontory, 3.25 m higher than the plot, giving unobstructed views of all birds and nest sites in the plot. At night, dawn, and dusk the density of boobies in the plot was high, confining birds to individual nest sites in pairs or alone. During daylight hours most birds leave the colony to forage; the remaining males and females have more freedom of movement, although mostly within our plot. Nonetheless, most birds spent most of their time at a nest site while in the plot during the day, occasionally moving to visit other birds or nest sites or to gather nest material. Most males had a single nest site that they occupied and defended during day and night, but males may visit other nest sites when those are undefended during the day, displaying to females and interacting with them there.

### Copulation behavior

We studied early-season breeding behavior in the plot from 23 October 2001–4 January 2002. The plot had approximately 70 potential nest sites; 41 females (hereafter, "breeding females") laid their clutches there during our behavioral observations. An additional 10 pairs began a clutch before observations started. Most of the remaining sites were defended by males that did not acquire a mate that season. During nightly leg-banding efforts in May and June 2001, the end of the previous breeding season, we banded all males and females encountered in and near the plot with either black (males) or

gray (females) numbered, plastic, field-readable leg bands. We continued this nightly banding throughout the behavioral study, banding the remaining birds when they were encountered. Many birds in the plot already had a numbered metal leg band, and we placed the plastic band on the unbanded leg in these cases. Any unbanded bird received both a plastic and a metal band, one band per leg. Due to the male-biased adult sex ratio, our sample has more males than females.

During daylight hours and adjacent dawn and dusk (0500–1830h local time) one observer documented all sexual interactions — including voluntary, forced, and rejected copulations — of adults in the plot using an "all-occurrences" sampling approach [33,34]. Copulations occur only on or near established nest sites in breeding colonies ( [35]; pers. obs.); in particular, the colony has no "hidden lek" area promoting between-sex interactions (i.e., [8]). Sexual events were classified as enumerated in Table 1. Except where noted, we combined the first three types and considered them "copulations", which we assumed represented voluntary participation by both male and female in copulation. In the minority of events in which the band numbers were not visible using a 10 x 42 binocular from the observation post, we left the post and moved unobtrusively to a better, temporary, position. Descriptive data were recorded in field notebooks and later transferred to digital files for analysis. Except for most of the period November 21–22, when an emergency prevented data collection, we monitored the plot throughout the local daylight hours and adjacent twilight periods (0500h–1700h). We supplemented the ambient light during the last 30 min of each day with an 8-watt fluorescent light located in the plot to facilitate band reading, in part to habituate the birds to the light for night observations (see below).

A total of five observers contributed to the all-occurrences behavioral sampling, with one primary observer (PME) conducting 60.3% of the total daylight observations. To estimate the inter-observer consistency, we conducted 15 hours of double-blind, simultaneous observations by the primary observer and one other observer.

To estimate frequency of copulation at night, we conducted all-occurrences sampling throughout three nights (Nov. 15 and 30, and Dec. 29). Ambient lighting was low during the new moon on Nov. 15, and on that night the fluorescent light in the plot remained on throughout the night. The artificial light, to which the birds had become accustomed from nightly exposure, had no discernible effect on the birds. The other two nocturnal samples were collected during full moons, with sufficient ambient light to see all birds in the plot clearly. Headlamps were used to read bands when necessary for observations during the first hour of daylight and during the night watches. The birds were generally indifferent to these activities.

Following [36], we consider extra-pair copulations to be "a special case of promiscuity where copulations occur with more mates than are included in the social mating system". We define a female's "social mate" as the male that shared incubation of that female's clutch of eggs. Copulations outside the pair of social mates could occur before a pair bond for the breeding season is even formed. Most studies of extra-pair sex in birds are silent regarding when a pair bond is considered to start, and specifying one moment for the creation of a pair bond may impose an arbitrary binary category

**Table 1. Categories of sexual events recorded in this study. Types 1, 2, and 3 are collectively considered "copulations".**

| Category | Behavior observed | n (% of total) |
|---|---|---|
| Type 1 | confirmed copulation; the observer saw full cloacal contact | 1010 (62%) |
| Type 2 | probable copulation; cloacal contact probably occurred, but was not fully visible due to the observer's angle of view | 297 (18%) |
| Type 3 | attempted copulation; both birds were positioned voluntarily and approximately correctly and the male bent his tail around the female's, but the cloacae did not touch or were judged not to touch | 249 (15%) |
| Type 4 | unreceptive female events; the male mounted the female but the female repelled the male, and/or did not adopt the appropriate posture, and/or did not open her cloaca | 82 (5%) |

on a fluid system of evaluation and choice. We do not know when individual pair bonds start in Nazca boobies, but we can specify unambiguously which male shares parental effort with a female. Copulations with additional males are called EPCs in this study.

## Nest histories, blood sampling, and parentage analyses

Nests in the plot were checked every day in the morning to record the laying and hatching dates of each egg, dates and causes of egg loss, and the band numbers of incubating and brooding adults.

Blood samples (< 50 µL) were taken by brachial or tibiotarsal venipuncture from the offspring in the nests under observation in the 2001–2002 breeding season on the day of hatching for molecular parentage analysis. The nestlings' social parents were sampled either immediately after their clutch or brood failed, or in March 2002 when their nestlings were approximately two months old. All non-breeding adults in the plot were also sampled in March to identify any extra-pair genetic parents. Blood samples (up to 400 µL) were stored in lysis buffer [37] at ambient temperature, and later at 4° C until analysis.

We used a standard multilocus minisatellite DNA fingerprinting protocol [38,39] to analyze the parentage of 22 nestlings sampled from 17 complete families and two additional chicks matched with only their social fathers. Nuclear DNA was extracted following a standard phenol/chloroform:isoamyl alcohol extraction protocol [40]. Aliquots of genomic DNA were digested with an excess of the restriction endonuclease *Hae*III, followed by fragment separation on agarose gels. Samples were arranged in family groups on gels to avoid scoring errors [41]. After electrophoresis, fragments were transferred to nylon membranes (Hybond N+, Amersham Biosciences) via Southern blot and hybridized to Jeffreys' probe 33.6 [38,39]. On one gel we had too few readable bands in some lanes using *Hae*III, so we evaluated the utility of an alternative restriction enzyme (*Hin*FI) by generating a second gel with this protocol but with *Hin*FI and Jeffreys' probe 33.6. The band-sharing values for the 18 readable dyads from digestion with *Hae*III were strongly correlated with those from *Hin*FI (Pearson's r = 0.74, P < 0.05, n = 18), so we treated the *Hin*F1 band-sharing data for the nine unreadable dyads on the uninformative gel as equivalent to the *Hae*III data.

To assess the parentage of each nestling, we tallied the number of offspring bands that could not be attributed to the fingerprints of either social parent and evaluated the proportion of bands nestlings shared with each putative parent as $2N_{AB}/(2N_{AB} + N_A + N_B)$, where $N_{AB}$ is the number of bands shared by the pair (dyad) of individuals of interest, $N_A$ is the number of bands unique to individual A, and $N_B$ is the number of bands unique to B [42,43]. These values were used to establish our criteria for determining extra-pair parentage.

First, we set the upper 95% confidence limit of the distribution of band-sharing for unrelated individuals as a cutoff value for parental exclusions [41]. We calculated this value separately for unrelated dyads (n = 37; 0.69) and nestling/unrelated adult dyads (n = 20; 0.59) because it could be inflated by high relatedness between mates due to extreme natal and breeding philopatry [32]. We used the distribution of unattributable bands among mother/nestling pairs to calculate the Poisson probability that mismatched bands were due to mutation and not extra-pair parentage [44]; this value was 0.004. Thus, Nazca booby nestling/parent dyads with two or more unattributable bands and band-sharing values falling below our cutoff levels were classified as extra-pair parentage.

We also calculated the probability of paternity mis-assignment, $X^P$, and the probability of mis-assigning an uncle as the father, $s^P$, as measures of error in our assessments of parentage. X is the mean band-sharing value for unrelated males (tabulated from 11 male-male dyads), P is the expected number of exclusively paternal bands, and s is the proportion of bands shared by siblings [44].

## Data analysis

Copulation can lead to fertilization, presumably motivating at least some of a female's overall copulation behavior. In Nazca boobies, fertilization of an ovum cannot occur outside the 4-day window preceding the egg's laying [26] and is

assumed to occur in the last ~ 24 h of this window (e.g., [45–47]). Acknowledging this, we anchored the copulation history of each breeding female in the plot to her own clutch initiation date; it is "day 0", the previous day is "day -1", and so on. Comparisons of the schedules of WPCs and EPCs were impeded by temporal variability in daily copulation behavior, due to the regular multi-day absences of females on pelagic foraging trips and probably to absences of a female's intended copulatory partners, on their own trips, when females were present. Thus, many bird-days with no copulations were structural zeroes (due to absence from the colony), but which zeroes are voluntary abstinence and which are structural is unknown. Given that absences during these weeks preceding laying tend to be ~ 2 d [27], and that the absences of at least two parties are involved in structural zeroes, we summarized individual copulation histories in 4-day blocks preceding each female's laying date (Fig 4). This binning smoothed the effect of colony attendance on copulation history.

To compare the schedules of WPCs and EPCs in the weeks preceding egg laying, we exploited the paired design with each female having both a WPC history and an EPC history in each 4-d block. Accordingly, we compared the schedules of WPCs and EPCs using a Wilcoxon Matched Pairs test (R package *rstatix*) for each 4-day block. The False Discovery correction for multiple comparisons [48,49] was then applied to the *P*-values. The times of day of WPCs and EPCs, summed within 1-h bins, were compared using the *chisq-test* R function. Statistical analyses were performed in R v. 4.4.1 [50].

The effect of a female's copulatory history on egg abandonment was evaluated using the number of WPCs and number of EPCs in the final four 4-day blocks (that is, 16 d) before the clutch was initiated. This period will be relevant to a male's assessment of his paternity risk. These values for copulation activity were used to predict the binary response variable "nest abandonment". Most nest failures during incubation result from the incubating parent leaving the colony when its mate is not present (unpub. data). We categorized breeding attempts as probable abandonments (hereafter, "nest abandonment") when the clutch was left unattended during the normal incubation period (43 d [51]); all such clutches disappeared, probably taken by Galápagos mockingbird (*Mimus macdonaldi*) egg predators, within a day of abandonment. Logistic regression was used to model the effects of EPCs and WPCs during these 16 d on the probability of nest abandonment, using the *glm* R package with a binomial link. First, a complex model including each predictor and their interaction was compared with a simple additive model with only the two predictors (i.e., number of EPCs and WPCs), using the analysis of deviance (the difference in model deviances is distributed as a Chi-square) provided by the package. Only the complex model could detect an effect of the ratio of WPC and EPC frequencies on probability of abandonment (for example, frequent abandonment when a male's mate had few WPCs and many EPCs) if an effect were present. Finding that the simple model was not significantly different from the complex model (*P* > 0.20), we used the simple model for inference.

## Results

### Validation of behavioral methods

During double-blind observations by two observers, we recorded a total of 21 mating events (all four Types combined); 18 events were detected by both observers, and three were detected by only one observer. During the three 24-hr periods with continuous observation, nocturnal events represented 1/32 (0.03), 1/27 (0.04), and 0/9 respectively of all mating events during the three double-blind observation periods (combined frequency = 0.03, 95% CI 0.00–0.10). Both nocturnal events were WPCs.

During 620 person-hrs of daily fieldwork during this study in colony areas adjacent to the plot, three plastic-banded adults normally seen in the plot were each recorded outside the plot on one occasion during our observation period in 2001–02. None of these was a female that bred in the plot. One was a non-breeding female seen frequently in the plot, one was a male that bred in the plot, and one was a non-breeding male. None of these birds was copulating or courting when observed outside the plot. For comparison, we documented 28,319 bird-hrs of presence of plastic-banded birds within the plot during the study. Thus, sexual activity appears to be restricted to the local area in which the bird will breed that season.

## Copulation patterns

We recorded a total of 1,638 sexual encounters during our 74-day study (Table 1). All of these encounters were a male mounting a female; none were a female mounting a male [52], as occurs in the congener brown booby *S. leucogaster* [53]. In a minority of cases (82, or 5.0% of the events), a male attempted to copulate with an uncooperative female, as indicated by her failure to position her tail and cloaca correctly after mounting and in some cases twisting her neck to repel the male with her bill or changing position to dislodge him. None of these 82 Type 4 events proceeded to a copulation. In only one of these cases did a male persist after a female resisted. In this unique case, a male attempted to copulate with a female resisting aggressively. This male repeatedly approached an incubating female, and the female responded aggressively but was constrained in her aggression by remaining in incubation posture. Even under this constraint the female prevented the male from mounting her after he placed one foot on the female's back.

The remaining 1,556 copulations or attempted copulations (our Types 1, 2, and 3; as a group, "copulations") involved a total of 56 females and 73 identifiable males; 17 copulations, almost all by the same female, involved an unknown number of unbanded, apparently itinerant, males. The female that copulated more than once with an unbanded (unrecognizable) male had an unknown number of partners; the remaining 55 females had 1–16 copulatory partners (median = 2). Most females copulated at least nine times during our observations (Fig 1). EPCs were common, representing 23.1% of the 1,276 copulations by breeding females. The most active female copulated 98 times, all with her social male; the most active non-breeding female copulated 88 times. We saw no evidence of females expelling semen after copulation. Copulations were more likely to occur just after dawn and just before dusk than during the intervening daylight hours (S1 Fig), corresponding to the higher colony attendance in these parts of the daylight period. Frequencies of WPCs and EPCs each peaked early and late in the day, but EPCs tended toward a more uniform distribution across the daylight hours (comparison of WPC and EPC frequencies by time period, $\chi^2 = 28.1$, df = 13, P < 0.01; S1 Fig).

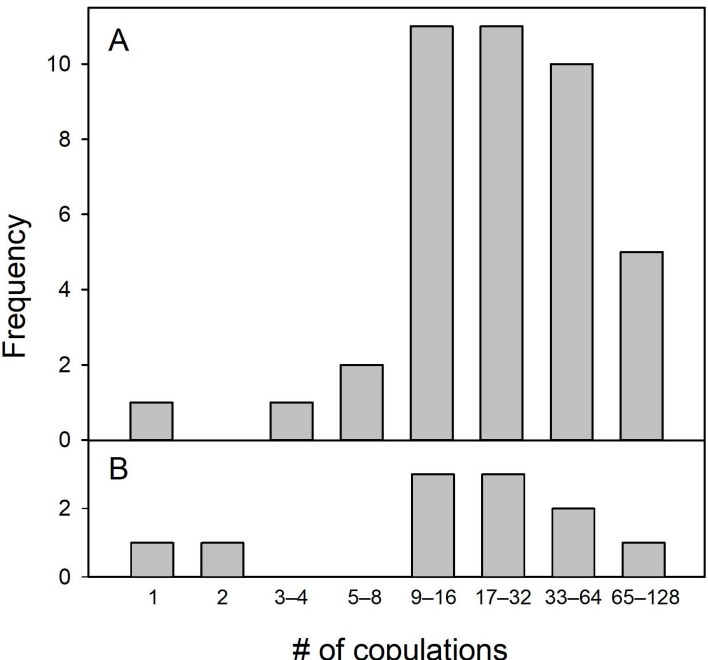

**Fig 1. Total numbers of copulations accumulated during our observation period by female Nazca boobies in the plot that copulated at least once (*n* = 56).** A: females that bred in the 2001–02 season. B: non-breeding females in 2001–02.

## Frequency of EPP

Band-sharing values among first-order relatives (mother-nestling, $n=22$ nestling-parent pairs) ranged from 0.64 to 0.90 and the distribution of social father-nestling values ranged from 0.65 to 0.87 (Fig 2); the two distributions were not statistically significantly different from each other (Mann-Whitney U, $Z=-1.45$, $P>0.10$; Fig 2). Both social parent/nestling band-sharing distributions did differ from the band-sharing distribution of all dyads of unrelated individuals (Mann-Whitney U-tests, $N_{mothers}=22$, $N_{fathers}=24$, $N_{unrel}=37$, $Z>6.2$ and $P<0.01$ in both cases; Fig 2) and from the distribution of nestling/unrelated adult values (Mann-Whitney U-tests, $N_{unrel}=20$, $Z>5.5$ and $P<0.01$ in both cases; Fig 2). Six father-offspring and two mother-offspring dyad band-sharing values fell just below the combined unrelateds cutoff value of 0.69, while no parent-offspring band-sharing value fell below the nestling/unrelated adult cutoff value of 0.59. Both father-nestling band-sharing values from incomplete families were well above both threshold values. Given these band-sharing data, the probability of mis-assigning an unrelated male as father was 0.003 and the probability of mistakenly assigning an uncle as father was 0.088.

Of the 22 nestlings from complete families, two had one unattributable band while the rest had zero. The offspring-parent band-sharing values for each parent for one of these two offspring and for the father of the other fell slightly below the unrelateds cutoff value; all band-sharing values were above the nestling/unrelated adult cutoff value. The average number of mismatched bands per offspring was 0.09 and a mean of 18.2 bands were scored per lane, yielding an estimated muta-tion rate of 0.09/18.2 or 0.005 mutations/locus/meiotic event. This value is comparable to that reported for other long-lived seabirds (e.g., 0.008 [54]; 0.004 [10]). Given these rates and the levels of band-sharing described above, we conclude that mutation can account for the two offspring with one unattributable band and that extra-pair parentage is not evident in this sample. The binomial 95% CI of the estimate of zero EPFs among the 22 nestlings was 0–0.15. Combining these results with those from the 32 nestlings from our earlier study [25], the binomial 95% CI of the estimate of 0 EPFs for Nazca boo-bies is 0–0.07 ($n=54$ nestlings). At least 18 of these nestlings were from second-laid eggs in two-egg clutches.

## To what degree do females control copulations?

None of the 1,556 copulations (Types 1, 2, and 3) appeared to be forced on the female by the male, inasmuch as the female did not attempt to move away from the male, or fight with the male, before, during, or after the event. In all of

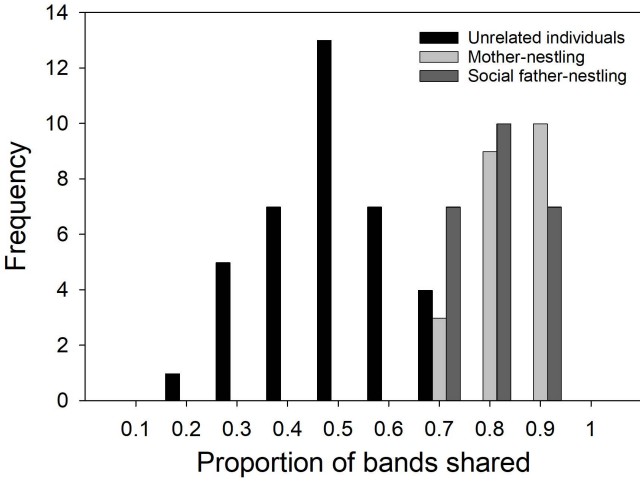

**Fig 2. Distribution of proportions of DNA minisatellite bands shared between pairs of unrelated individuals (n =37), mothers and nestlings (n=22), and social fathers and nestlings (n=24).** The majority of parent-nestling dyad values fell above the upper limit for unrelated pairs (0.69) and all nestling-parent comparisons were above the nestling-unrelated individual upper limit (0.59) indicating low probabilities of misassigning unrelated males as a parent.

these cases the female appeared to facilitate the copulation by positioning her tail correctly after the male mounted her. In almost all cases, the male and female were actively courting immediately before the copulation, including sky point displays by the male and handling of nest material [35], or had been actively courting in the previous hours. In the single exception, a male landed in the plot and immediately attempted (unsuccessfully) to copulate with his social mate for that season without prior courtship.

A breeding female could demonstrate the intention to participate in an EPC by moving from her eventual social male's nest site to another male's location to copulate. Birds of each sex were able move around the plot during daylight. In 132 of the 359 EPCs, the female occupied a single home nest site around the date of the copulation; 19 of these 21 females eventually laid her clutch at that site. These females each walked from that current home site to the location of the male with which she had the copulation in 81/132 (0.61, range 1–18 m; Fig 3) of these cases. For 124 of these 132 EPCs, the male at the current home site could be assessed unambiguously as present or absent at his nest site from our data; in 31 of these cases (0.25), that male was present and almost certainly observed the EPC, given the short sight lines involved in most events. Females made no discernible attempt to conceal EPCs. In no case did the home site male leave his nest site during the female's "foray" [55] away from that site or make other discernible attempt to mate-guard. In the subsequent 3 hrs., or until the end of the observation day, the female returned to the home site male in 22 of these cases after 2–161 min. (median = 8.5 min.). In none of these cases did the male exhibit any aggressive or other retaliatory behavior. In all 22 cases, that male performed pair-bonding behaviors (courtship displays, allopreening, copulation, resting together) to the female when she returned.

### Schedules of WPCs and EPCs in breeding females before egg laying

Of the 56 females involved in copulations, 41 initiated a clutch during our 74-day behavioral study. For these 41 breeding females, copulations preceding their first egg (their "pre-laying window") are relevant to fertilization of that egg; other copulations occurred between first and second eggs, and after a clutch's failure. Breeding females had 1–11 copulatory partners (median = 2) in their individual pre-laying window, and 25 (0.61) had two or more copulatory partners. Females accumulate copulations over time during their pre-laying window: for breeding females, the number of days of observation in their pre-laying window predicted number of copulations during that window (linear regression $F_{1,39} = 20.00$, $P < 10^{-4}$, $r^2 = 0.32$, slope = 0.58). The number of days of observation also predicted the number of copulatory partners in

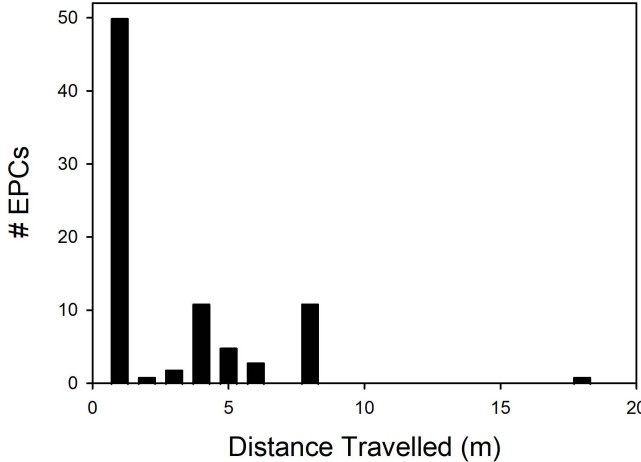

**Fig 3. Distances travelled from her home nest site by female Nazca boobies ($n = 21$) before having an EPC.**

the pre-laying window ($F_{1,38} = 21.07$, $P < 0.0001$, $r^2 = 0.34$, slope = 0.60). Thus, the proportion of females with more than one partner and the median number of copulatory partners may be underestimated for females that began copulating before our observations began. Considering only the 27 females with pre-laying observation windows ≥ 20 days long (median window length of these = 40 days), 19 of these (0.70) had two or more copulatory partners before initiating a clutch (median = 2). In summary, a large majority of breeding females had multiple copulatory partners.

All 41 breeding females had at least one WPC, and they copulated more with their eventual social mates (981/1276 occasions; 0.77), despite the fact that most had more than one copulatory partner. WPCs and EPCs each occurred well before laying the first egg in a clutch (Fig 4), and well before the 4-d period preceding laying representing part or all of the fertilization window of this species [26]. WPC rate increased as the egg-laying day approached, and almost all 41 females had at least one WPC in each of the last two 4-d blocks preceding laying (Fig 4). The median number of EPCs per 4-d block never exceeded zero; nonetheless, EPCs were common until 4 days before laying, when EPCs were almost absent and WPC rate had increased rapidly (Fig 4). Per capita WPC rate exceeded EPC rate to a statistically significant degree, after correcting for multiple comparisons, only in the last four 4-d blocks preceding laying (S1a Table).

Most females contributed partial copulatory histories: all histories ended on the female's day −1, but early layers had unobserved early blocks that occurred before our observations began. Thus, some females contributed to all blocks in our analysis, and others contributed only to later blocks. To address possible non-independence within individuals across blocks that might affect the temporal outcome, because females had different representations in the different blocks, we repeated the analysis using only the smaller sample of females that had complete copulatory histories from the −28 d block through the −4 d block. This standardized the contribution of females to any temporal trend. The result was similar: per capita WPC rate exceeded EPC rate to a statistically significant degree only in the last two 4-d blocks preceding laying (S1b Table).

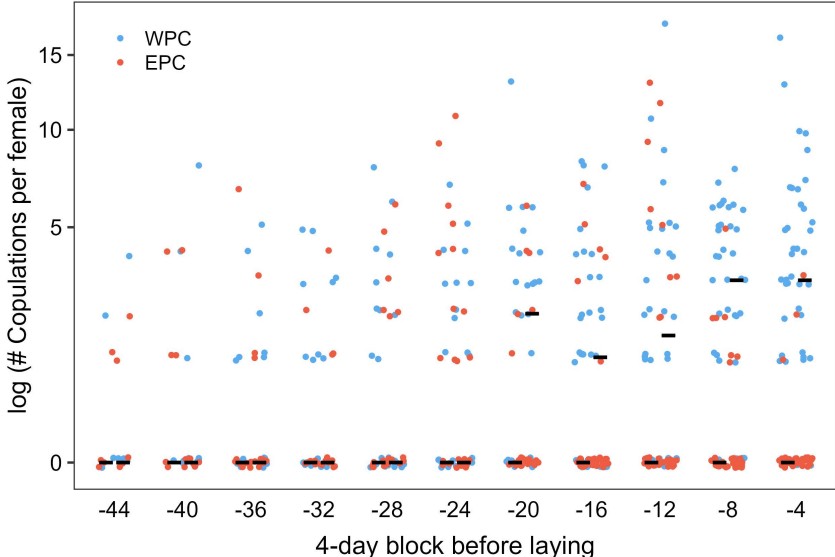

**Fig 4. Temporal patterns of breeding female Nazca boobies' EPCs and WPCs, binned in 4-day blocks and standardized to each female's laying date to start a clutch of eggs.** Each female has one EPC point and one WPC point in each block. Note log scale on Y axis, to enhance resolution at smaller Y values. Median numbers, across the sample of females, of EPCs and WPCs per block are shown by pairs of black horizontal lines, with EPC median the left of the pair and WPC median the right. WPC medians are > 0 in the last five blocks before egg-laying. See text for within-block statistical significance tests of differences between distributions of EPCs and WPCs.

Of 294 female-blocks analyzed, 80 contained no copulations at all, perhaps due to a female's, or a copulatory partner's, absence. Using only the 214 female-blocks with ≥ 1 copulation, WPC rate again exceeded EPC rate to a statistically significant degree only in the final four 4-d blocks before laying (S1c Table).

A minority of breeding females never had an EPC and always contributed zeroes to each 4-d block. The "no-EPC" females may represent a distinct subpopulation (lacking motivation or opportunity for EPCs) among breeders from the majority "some-EPC" females, and excluding them clarifies the pattern for "some-EPC" females. For the "some-EPC" females, the WPC rate exceeded the EPC rate significantly only in the last two 4-d blocks (S1d Table). In summary, breeding females engage in EPCs regularly before starting their clutch, even more than a month before laying, but both decrease EPC rate and increase WPC rate in the 8 days before laying, or up to 16 days before, depending on analytical context.

Females may add a second egg to the clutch in the eight days after laying the first egg [29], the second egg improving reproductive success through its insurance value [28], mitigating the exceptionally poor hatching success in this species [56]. We expected EPCs to be common between the first and second eggs: insurance sperm from a different male might improve hatching success, and social males were typically absent on foraging trips during a female's incubation stints. During these eight days we recorded 92 copulations; only six (0.07) were EPCs.

### Male post-laying behavior and EPC

Loss of EPF offspring before hatching could mask EPF frequency when paternity analysis is done for hatchlings, as we have done. Using the 33 females with complete copulatory histories for the 16 d. before egg laying, we used daily nest histories to identify cases of early abandonment. Eighteen of these breeding attempts were abandoned and the egg(s) depredated, and 15 hatched an egg at the expected time. Of the 18 abandonments, five occurred after the end of the typical incubation period (range 69–103 d after egg laying), and we do not consider them probable responses to uncertain paternity, leaving 13 "premature" abandonments. Considering the remaining 13 nest abandonments and 15 nests without abandonment, logistic regression did not support an effect of either number of EPCs in the last 16 d ($P > 0.10$) or number of WPCs in the last 16 d ($P > 0.10$; S2 Table) on abandonment. Furthermore, in 38 years of intensive monitoring of breeding in this colony we have never observed a male (or female) eject an egg from the nest, as is conspicuous in blue-footed booby males when paternity is in doubt [57].

### Discussion

Genetic evidence shows that many female birds are inseminated by extra-pair males [1]; often, at least some of this extra-pair activity seems to be voluntary by the females and opposed by her social mate [55]. The behavioral link connecting female motivation to obtain EPCs to the genetic outcomes in her offspring is largely lacking, because the females often hide EPCs from social males, and consequently also from observers. Intensive effort like radio-tagging (e.g., [58]) or experimental manipulations (e.g., [59]) may help, but even then the inference that an EPC has occurred is often indirect, rather than an actual observation of EPC (e.g., [60]) or occurs under artificial circumstances. Crowded colonial species that copulate within the colony, like Nazca boobies, may relieve researchers of the observability problem to some extent. We used a large sample of sexual behavior in a visually open colony to evaluate the relative degrees of control by females vs. males in copulation. Female Nazca boobies were wholly unconstrained by males in their sexual activities, and most copulated with multiple males before starting their clutch. All copulations were voluntary. EPCs frequently occurred a short walk from the nest site that a female shared with her eventual breeding partner, sometimes in full view of that male, and he welcomed her back to his own nest site when she finished. Males did not mate-guard. We found no evidence of post-hatching costs imposed on females following EPC activity: clutch abandonment was unrelated to the mother's EPC frequency, and males cannot discriminate among hatched brood-mates because the brood is effectively one nestling [29,30]. Thus, females controlled their copulatory activity and apparently incurred no cost from their breeding partner

for having EPCs. To our knowledge, this is the first robust evidence of complete – even unfettered – sexual agency in a female bird aside from lek mating systems.

The promiscuity of female Nazca boobies often involved initiative by the female (walking to a male's nest site), and thus evidently reflects motivation on the female's part. This opportunity to document the degree of motivation in unconstrained females, in a species without any underlying fitness benefit involving paternity, may provide an approximation of the lower bound of female birds' motivation to engage in EPCs. The large majority of bird species studied *do* have EPFs [1], and so those females may have additional incentive to copulate promiscuously (we acknowledge that forced copulation (e.g., [61]) can also contribute to a female's copulatory history in other species). This perspective contributes to the substantial literature on sexual conflict, ranging from Nazca boobies in which females apparently win the conflict, to the often-covert character of female forays for EPCs in other species, including nocturnal forays in normally diurnal birds [62], when females are more encumbered by males.

Male birds in nominally monogamous bird species are expected to oppose EPCs by their mate when it reduces their paternity in their joint offspring, presages a divorce, increases their exposure to sexually transmitted diseases, or otherwise compromises their fitness (e.g., [63]). Mate-guarding and other countermeasures by male birds in many species support this prediction [64]. In contrast, the sexual agency of Nazca booby females results from an atypical absence of any male coercion; this circumstance has two possible, additive, explanations. First, EPFs were absent in this study and previously [25]; thus, coercion was unnecessary to guard paternity. However, male Nazca boobies may incur other costs of EPCs by their mates. Second, the social context for sexual interactions of Nazca boobies is unusual, representing one end of a spectrum for monogamous birds and limiting the potential for coercion to yield a positive fitness outcome for a male: females are the strongly limiting sex [16,23,65], representing a valuable resource that males can ill afford to alienate by coercion or divorce. Aside from these considerations of fitness consequences, females' substantial advantage in physical power probably would make attempts to coerce them ineffective, representing a constraint on male options. Males probably cannot "police" females in this species and apparently would not obtain a paternity benefit from it anyway.

EPCs were common, but EPF frequency was estimated as zero and is certainly rare if it ever occurs; this pattern has been observed in some other species (e.g., American kestrels *Falco sparverius* [66]). The temporal pattern of copulation behavior illuminates this outcome. EPC frequency suddenly approached zero approximately 4 d. before clutch initiation, simultaneous with higher WPC frequency (Fig 4). For females with at least one EPC during our study, WPC frequency exceeded EPC frequency to a statistically significant degree only in the last approximately 8 d. before egg laying, but did so markedly (Fig 4, S1c Table). Because Nazca booby females ovulate 4 d. before clutch initiation [26]), ova can be fertilized only during this brief window, and females focus their copulatory activity into WPCs then. Several aspects of avian copulation should give social males an advantage in sperm competition and resulting paternity in this situation, including the "last-in, first-out" stratification in sperm storage tubules, dilution of EPC sperm, and time-dependent passive loss of sperm [67]. Additionally, males may strategically allocate fewer sperm to EPCs if the female's objective in the interaction is not fertilization of an ovum [68]. Thus, males are incented to attend the colony as much as possible to overlap in presence with their intended social mate, to take advantage of WPC opportunities that she provides. Male Nazca boobies do attend the colony increasingly as the date of clutch initiation approaches, and consistently attend more than females during this time, with evidence of a physiological cost incurred from this extra attendance [27].

In seabirds, the separation of the feeding site from the breeding site may deprive (or relieve) a female of her future breeding partner on some copulatory occasions, due to his absence on a feeding trip, with alternative copulatory partners readily available in the colony [69]. This potentiates EPCs, and many occurred during the absence of the social male in this study; indeed, females tended to engage in EPCs during daylight hours, when colony attendance is lowest in this species [70], and not at dawn and dusk, when WPCs were concentrated (S1 Fig). Other EPCs were known, visually, to the female's male partner, but they simply accepted the female extra-pair behavior. With male coercion removed from female sexual activity, this activity can evolve under a simpler set of costs and benefits associated with EPCs than in

many birds. We do not analyze these possible benefits and costs, other than essentially excluding any benefit requiring fertilization. Thus, hypotheses involving fertilization insurance, or females manipulating EPC males into favoritism toward her impending family (e.g., allofeeding her or her offspring) based on paternity, seem to be moot. More plausibly, females might be assessing potential mates via EPC before making a final decision on this season's father. Likewise, the serial monogamy mating system of Nazca boobies, involving frequent mate changes between breeding attempts [16], suggests a possible social benefit of maintaining relationships with past or future breeding partners (e.g., [71]) in this long-lived species; colloquially speaking, female Nazca boobies keep a "little black book" of males under this hypothesis, with sex part of relationship-building. Females might acquire beneficial (or harmful) microbes during EPCs by cloacal inoculation [72], casting EPCs as sex but not procreation. Finally, EPCs by females could be an epiphenomenon of selection on males to acquire EPCs [73]. We found that females apparently have complete sexual agency, equivalent to that of some females in leks [5], with no covert or forced copulations; this will facilitate future evaluation of the evolution of sexual behavior in Nazca boobies.

## Supporting information

**S1 Fig. Variation in frequencies of (A) WPCs of breeding females, (B) EPCs of breeding females, and (C) copulations of nonbreeding females with time of day, during daylight and crepuscular hours.** Copulations outside of these hours are rare (see main text).
(PNG)

**S1 Table. Comparisons of WPCs and EPCs accumulated by breeding females, aggregated by 4-d time block preceding the start of a clutch, for 1a) all available time blocks, 1b) only females that had complete copulatory histories from blocks −28 d through −4 d, 1c) only time blocks in which a given female copulated at least once, and 1d) only females that had at least one EPC at any point before starting a clutch.** 4-d blocks are named by the first day in the block. By performing the same statistical test for each year, we increased the probability of falsely rejecting a null hypothesis. We used the false discovery rate procedure [48], which controls the expected fraction of null hypotheses that are mistakenly rejected, to appropriately adjust observed P values for this problem. The procedure ranks the $n$ comparisons in order of decreasing $P$ values and compares $P$ values to a critical significance level, beginning with the largest $P$ value. The critical significance level for each comparison, $d_i$, is calculated by dividing the specific comparison i by the total number of comparisons n and then multiplying by the false discovery rate (the expected proportion of null hypotheses mistakenly rejected). For example, in Table 1a, the fifth comparison (the comparison with the fifth largest $P$ value) of 11 total comparisons, given a false discovery rate of 0.05, has a $d_i$ of 0.023 (= 5/11 x 0.05). The achieved significance level (0.08) is > $d_i$ for that comparison, so the null hypothesis is not rejected for that comparison [48,49]. For the fourth comparison, the achieved significance level of 0.007 is less than the fourth $d_i$ (0.018), so the null hypothesis is rejected for that comparison and for all subsequent comparisons [48,49]. Rejections of null hypothesis are indicated by bold font.
(DOCX)

**S2 Table. Logistic regression evaluating effects of WPC and EPC histories of females on the probability of "premature" abandonment of clutches after laying (see main text).**
(DOCX)

## Acknowledgments

We thank the Galápagos National Park Directorate for permission and support to work in the Park, and the Charles Darwin Research Station and TAME airline for logistical support; T. Ursillo for statistical input, B. Tague for material assistance with lab work, P. Dunn for laboratory protocol advice, J. Quinn who kindly provided Jeffreys probes, and the academic editor and reviewer D. Wysocki for comments on a previous draft.

## Author contributions

**Conceptualization:** David J. Anderson.

**Data curation:** David J. Anderson, Kathryn P. Huyvaert, Paola M. Espinosa.

**Formal analysis:** David J. Anderson, Kathryn P. Huyvaert, Paola M. Espinosa.

**Funding acquisition:** David J. Anderson.

**Investigation:** David J. Anderson, Kathryn P. Huyvaert, Paola M. Espinosa, Mark A. Westbrock, Jill A. Awkerman, Eric G. Schneider.

**Methodology:** David J. Anderson, Paola M. Espinosa.

**Project administration:** David J. Anderson.

**Resources:** David J. Anderson.

**Supervision:** David J. Anderson.

**Validation:** David J. Anderson.

**Visualization:** David J. Anderson, Kathryn P. Huyvaert.

**Writing – original draft:** David J. Anderson, Kathryn P. Huyvaert, Paola M. Espinosa.

**Writing – review & editing:** David J. Anderson, Kathryn P. Huyvaert, Mark A. Westbrock, Jill A. Awkerman, Eric G. Schneider.

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
