## [Decision Letter · Decision Letter 0]

9 Aug 2025

Dear Dr. Anderson,

Thank you for submitting your manuscript to PLOS ONE. After careful consideration, I feel that it has merit but still requires a few minor clarifications to fully meet PLOS ONE’s publication criteria. Therefore, I invite you to submit a minor revision of the manuscript that addresses the points raised during the review process.

We look forward to receiving your revised manuscript.

Kind regards,

Steven E. Travis, PhD

Academic Editor

PLOS ONE

Journal Requirements:

2. To comply with PLOS ONE submissions requirements, in your Methods section, please provide additional information regarding the experiments involving animals and ensure you have included details on methods of anesthesia and/or analgesia.

5. Please remove all personal information, ensure that the data shared are in accordance with participant consent, and re-upload a fully anonymized data set.

Additional Editor Comments:

As you will see, the lone reviewer of your manuscript (no other reviews could be acquired in spite of over a dozen invitations) was very enthusiastic about the importance of this contribution to the literature on extra-pair copulation in birds, particularly considering the complete sexual agency demonstrated by females.  In fact, this reviewer recommended acceptance with no further changes.

As Academic Editor, I conducted my own independent review of the manuscript, with my comments appearing below.  Please pay careful attention to each of these comments in crafting your revisions.

Academic Editor comments:

Lines 119-120:  Did unpaired males simply roam the study plot or did they try to associate themselves with a nest?

Lines 179-180:  Please clarify what results were correlated, since an r of 0.74 wouldn’t be very reassuring if you’re referring to the assignment of parentage.

Lines 218-219:  How were times of day categorized?  Also, were multiple days for each female included in the analysis?  If so, some sort of comparison among females should first be conducted to rule out non-independence of observations within females.

Line 231:  by “the Chi-square test within ANOVA,” do you mean analysis of deviance using the deviance statististic, G^2^ , which is distributed as a Chi-square?

Lines 234-235:  This feels like a result more than a method.  I would suggest making this statement in the Results.

Lines 275-277:  It’s not entirely clear what this Chi-square test was evaluating.  Was it a comparison of the number of EPCs vs. WPCs taking place by time period?

Lines 327-328:  Were there EPCs observed for females that did not occupy a single home nest site around the date of the copulation?

Fig. 4:  The y-axis label should report the log-transformation.

Lines 437-440:  Is it fair to call it sexual conflict when the EPC behavior of females apparently results in no negative consequence to male fitness?

Reviewer's Responses to Questions

**Comments to the Author**

1. Is the manuscript technically sound, and do the data support the conclusions?

Reviewer #1: Yes

2. Has the statistical analysis been performed appropriately and rigorously?

Reviewer #1: Yes

3. Have the authors made all data underlying the findings in their manuscript fully available?

Reviewer #1: Yes

4. Is the manuscript presented in an intelligible fashion and written in standard English?

Reviewer #1: Yes

Reviewer #1: • What are the main claims of the paper and how significant are they for the discipline?

This paper is extremly important for behavioural ecology. This is the first robust evidence of complete sexual agency in a female bird aside from lek mating systems.

• Are the claims properly placed in the context of the previous literature? Have the authors treated the literature fairly?

I did not find any irregularities

• Do the data and analyses fully support the claims? If not, what other evidence is required?

The obtained results are so extraordinary that the authors took into account all possible sources of error. Therefore, I have no comments regarding the data and analyses.

• If the paper is considered unsuitable for publication in its present form, does the study itself show sufficient potential that the authors should be encouraged to resubmit a revised version?

I don't know how this article could be improved. The relationship of EPC to future mate choice should be a topic of future work by this team.

**Do you want your identity to be public for this peer review?** For information about this choice, including consent withdrawal, please see our Privacy Policy

Reviewer #1: **Yes: ** Dariusz Wysocki

---

## [Author Response · Author response to Decision Letter 1]

25 Sep 2025

It's in the Response to Reviewers documents

---

## [Editor Report · Decision Letter 1]

13 Oct 2025

Female sexual agency and frequent extra-pair copulations, but no extra-pair paternity, in Nazca boobies (Sula granti)

PONE-D-25-23666R1

Dear Dr. Anderson,

We’re pleased to inform you that your manuscript has been judged scientifically suitable for publication and will be formally accepted for publication once it meets all outstanding technical requirements.

Kind regards,

Steven E. Travis, PhD

Academic Editor

PLOS ONE

---

## [Editor Report · Acceptance letter]

PONE-D-25-23666R1

PLOS ONE

Dear Dr. Anderson,

I'm pleased to inform you that your manuscript has been deemed suitable for publication in PLOS ONE. Congratulations! Your manuscript is now being handed over to our production team.

Kind regards,

on behalf of

Dr. Steven E. Travis

Academic Editor

PLOS ONE